# Enhancing Oil Recovery by Polymeric Flooding with Purple Yam and Cassava Nanoparticles

**DOI:** 10.3390/molecules28124614

**Published:** 2023-06-07

**Authors:** Hasanain A. Al-Jaber, Agus Arsad, Muhammad Tahir, Mustafa Jawad Nuhma, Sulalit Bandyopadhyay, Abdulmunem R. Abdulmunem, Anis Farhana Abdul Rahman, Zakiah binti Harun, Augustine Agi

**Affiliations:** 1Institute for Oil and Gas, Faculty of Engineering, Universiti Teknologi Malaysia (UTM), Skudai 81310, Johor, Malaysia; 2Department of Chemical Industries Technologies, Southern Technical University, Basrah 61006, Iraq; 3UTM-MPRC Institute for Oil and Gas, Faculty of Engineering, Universiti Teknologi Malaysia (UTM), Skudai 81310, Johor, Malaysia; 4Chemical and Petroleum Engineering Department, United Arab Emirates University (UAEU), Al Ain P.O. Box 15551, United Arab Emirates; 5Chemical Engineering Department, College of Engineering, University of Al-Qadisiyah, Al-Diwaniyah City P.O. Box 88, Iraq; 6Department of Chemical Engineering, Norwegian University of Science and Technology, Høgskoleringen 1, 7491 Trondheim, Norway; 7Electromechanical Engineering Department, University of Technology-Iraq, Baghdad 10066, Iraq; 8School of Chemical and Energy Engineering, Faculty of Engineering, Universiti Teknologi Malaysia (UTM), Skudai 81310, Johor, Malaysia; 9Faculty of Chemical and Process Engineering Technology, College of Engineering Technology, Universiti Malaysia Pahang, Gambang 26300, Pahang, Malaysia; 10Centre for Research in Advanced Fluid and Processes (Fluid Centre), Universiti Malaysia Pahang, Gambang 26300, Pahang, Malaysia

**Keywords:** enhanced oil recovery, polymer flooding, nano-polymer, cassava nanoparticles, purple yam nanoparticles

## Abstract

Significant amounts of oil remain in the reservoir after primary and secondary operations, and to recover the remaining oil, enhanced oil recovery (EOR) can be applied as one of the feasible options remaining nowadays. In this study, new nano-polymeric materials have been prepared from purple yam and cassava starches. The yield of purple yam nanoparticles (PYNPs) was 85%, and that of cassava nanoparticles (CSNPs) was 90.53%. Synthesized materials were characterized through particle size distribution (PSA), Zeta potential distribution, Fourier transform infrared spectroscopy (FTIR), differential scanning calorimetry (DSC), and transmission electron microscopy (TEM). The performance of PYNPs in recovering oil was better than CSNPs, as found from the recovery experiments. Zeta potential distribution results confirmed the stability of PYNPs over CSNPs (−36.3 mV for PYNPs and −10.7 mV for CSNPs). The optimum concentration for these nanoparticles has been found from interfacial tension measurements and rheological properties, and it was 0.60 wt.% for PYNPs and 0.80 wt.% for CSNPs. A more incremental recovery (33.46%) was achieved for the polymer that contained PYNPs in comparison to the other nano-polymer (31.3%). This paves the way for a new technology for polymer flooding that may replace the conventional method, which depends on partially hydrolyzed polyacrylamide (HPAM).

## 1. Introduction

Due to the depletion of fossil resources and environmental challenges, biodegradable nanomaterials have received much attention in recent decades as new materials that can be involved in oil recovery [1]. These nanocrystal materials have gained outstanding properties in comparison to their counterparts, the microparticles, through their high surface-to-volume ratio, as they are rigid materials at the nanometer scale [2,3,4]. Recent studies have shown that these nanomaterials can be used as fillers to improve the mechanical and barrier properties of bio-composites [5]. For these industrial applications, a continuous endeavor is undertaken to find innovative solutions to achieve an efficient and sustainable performance for these industries. Therefore, starch nanoparticles have been the focus of a number of works that are devoted to developing the bio-composites by blending such starch nanoparticles into the biopolymeric matrices [6,7,8]. In fact, starch can be found in the stems, roots, fruits, and seeds of many plants, such as sweet potato, cassava, potato, and many more. In addition to its original form, starch can be modified by reducing its size. Starch nanoparticles have a small size and a large active surface area, making them suitable for using as fillers or as a reinforcing material in biopolymers.

Starches, being biodegradable natural polymers, are good candidates for the formation and production of nanoparticles. The market for starches is constantly growing, leading to a continuous search for products with specific features that meet industry requirements. The modification of starch with acid hydrolysis has been used to modify the structure of the granules and produce more soluble starch combinations [9]. Starch is usually hydrolyzed with mineral acids such as acetic acid to remove the amorphous regions and retain the crystals [10,11].

Being used in oil recovery is one of the creative applications of starch that has many challenges. Starch nanoparticles made from cassava and purple yam have received great interest because they are cheap, abundant, non-toxic, and biodegradable materials [12,13]. During the processing of cassava and purple yam tubers into starch, the tuber is peeled off and then subjected to a number of physical operations to extract the starch and convert it to nano-size [14]. There are many factors that affect the production of nanoparticles from starch. These factors include: the process of hydrolysis, the type of acid used, the concentration of the acid, the amylopectin-to-amylose ratio in starch, the concentration of starch, and the time, temperature, and speed of hydrolysis [15]. Additionally, the ultrasound technique has been widely used to produce granules of nano-sized materials. On the contrary, exposure of the polymer solution to the high intensity of ultrasound radiation may reduce the molar mass of the starch.

Despite that water flooding can recover significant amounts of oil, nonetheless, this amount cannot exceed 50% from OOIP in the best scenario [16,17]. On the other hand, the common expectation of polymer flooding is only 15% to 20% incremental recovery over secondary flooding using synthetic polymers such as HPAM [10]. Therefore, searching for natural polymer materials can help in reducing the operational costs, and at the same time, may lead to improvements in oil recovery. Nadia et al. [18] investigated the starch extracted from purple yam and evaluated its physicochemical and functional properties. For this purpose, five cultivars of yam were tested in their study. Scanning electron microscopy (SEM) was applied and three different kinds of granules, based on their shapes, were identified: round, oval, and spherical. Furthermore, they measured the relative crystallinity of the produced starch, which was found between 20.6% and 30.4% with the second yam type.

Cassava starch-grafted polyacrylamide (CASPAM) hydrogel was synthesized according to a method proposed by Matovanni et al. [19] by using the microwave technique and an initiator. The characterization of the produced CASPAM was examined by FTIR and SEM analyses. To predict the behavior of the samples under reservoir conditions, the properties of CASPAM, such as water-solubility and viscosity, were determined as a function of the temperature, salt concentration, and aging time. The FTIR spectra and SEM analysis for CASPAM confirmed that the polyacrylamide chains were successfully grafted onto the cassava starch radix. It was found that preparation of CASPAM with 10 g of acrylamide and 180 s of irradiation by use of a microwave resulted in obtaining a high grafting percentage and water solubility, which were 15.66% and 96.06%, respectively. The results from this study showed that CASPAM helped in improving the temperature resistance and durability to high salinity in relation to the observed reservoir conditions. This indicated that mixing cassava with polyacrylamide has good potential for oil recovery operations as an efficient, reliable, and economic technology. 

Based on previous research, the current work aimed to synthesize starch nanoparticles from both cassava and purple yam and integrate them with a constant concentration of HPAM (2000 ppm). This was achieved by involving each kind of starch nanoparticle in the polymer flooding with HPAM, and the new injection process led to the improvement of oil recovery. The optimum concentration for each kind of nano-starch was estimated from interfacial tension measurements and rheological properties at a temperature of 60 °C. This temperature was chosen to simulate the original temperature of Langgak oilfield in Sumatra, Indonesia. The experiments of polymer flooding were performed on a true crude oil from the oilfield and on core samples similar in their properties to that of the original field sandstone. The quality and properties of the synthesized nanoparticles were evaluated using PSA, TEM, Zeta potential distribution, FTIR, and DSC analyses.

## 2. Results and Discussion

### 2.1. TEM Analysis

The final weight of the produced nanoparticles was found to be 36.21 g for CSNPs and 34 g for PYNPs. Based on that, the yield of the produced nanoparticles for CAS and PYS was 90.53% and 85%, respectively. Figure 1 and Figure 2 show the particle size distribution and TEM analysis results for the samples of PYNPs and CSNPs, respectively. Figure 3 and Figure 4 show the TEM images for PYNPs and CSNPs, with some morphological details. These images demonstrate that the particles were well-distributed, with the particle size ranging from 5.4 to 13.8 nm for PYNPs (Figure 3a) and from 12.2 to 24 nm for CSNPs (Figure 4a). Large quantities of small particle sizes for PYNPs and CSNPs were produced during the preparation process of these nanoparticle components [20].

The average particle size for PYNPs and CSNPs based on the intensity percent (which is useful in detecting small amounts of aggregation) was 363.12 and 52.92 nm, respectively. The shape of the produced nanoparticles varied from spherical to hexagonal and rod-like, which it is shown in the TEM images. As Ku and Maynard [21] stated, the combinations of monopolar electric forces for these nanoparticles, which are controlled by the applied temperature, have well-contributed to the formation of a large portion of non-agglomerated and spherical nanoparticles.

As the coagulation between particles rapidly decreased due to quenching and dilution effects [22], the particles moved from a spherical shape to other forms, such as hexagonal and rod-like, as shown in Figure 3 and Figure 4. Free radicals were produced during the cavitation process, which involves the formation, growth, and collapse of bubbles between the particles [23]. The violent collapse of these cavitation bubbles is illustrated by red circles in Figure 3c and Figure 4c.

The morphology and surface appearance of the nanoparticles for both types were detected by TEM analysis. The micrographs revealed that both purple yam NPs (Figure 3a) and cassava NPs (Figure 4a) were approximately hexagonal in shape, with fairly smooth surfaces, and monodispersed with a uniform size. As mentioned before, the size of the PYNPs (363.12 nm) was bigger than that of the CSNPs (52.92 nm). The polydispersity index (PDI), in which the numerical value ranges from 0 for a perfectly uniform sample regarding the particle size, to 1 for a highly polydisperse sample with multiple particle size populations [24,25], is an important factor that contributes to the stability and homogeneous distribution of the produced nanoparticles. Its mean value was calculated for PYNPs as 0.937 and that for CSNPs as 0.916, based on the intensity percent analysis. Table 1 displays the hydrodynamic particle size, polydispersity index (PDI), and Zeta potential for PYNPs and CSNPs in the aqueous solution.

### 2.2. Zeta Potential Outputs

Three runs were processed for the Zeta potential analysis for PYNPs and CSNPs, as illustrated in Figure 5 and Figure 6, respectively. The mean Zeta potential for PYNPs was −43.9, −33.8, and −31.3 mV for the green, yellow, and red curves, respectively. This indicates that the stability of the particles was high enough that the nanoparticles did not aggregate [26,27]. This makes PYNPs suitable candidates for injection in the oilfield reservoirs. On the other hand, the mean Zeta potential for CSNPs obtained from the same instrument was −9.3, −10.3, and −12.4 mV for the same color combination (green, yellow, and red), as shown in Figure 6. This means that CSNPs were critically stable in water formation.

### 2.3. FTIR Formation Analysis

FTIR spectra for both PYNPs and CSNPs are shown in Figure 7. For CSNPs, the results of the FTIR analysis can be explained as follows: The peak value of 3371 cm^−1^ represents a medium N-H stretching bond, consisting mainly of aliphatic primary amines [28]. The peak of 2933 cm^−1^ refers to medium C-H stretching and indicates that the main compound within this absorption consisted of alkanes. The peak value of 2345 cm^−1^ indicates a strong O=C=O stretching bond, and it mainly consisted of carbon dioxide. The peak of 1648 cm^−1^ represents a strong C=O stretching bond with materials composed of the component lactam. Lactam (which is the combination of the words lactone + amide) is a cyclic amide and it is essentially derived from an amino alkanoic acid [29]. 

The value of 1158 cm^−1^ refers to a strong C=O stretching, and the main compound for this combination consisted of tertiary alcohol. The value of 1084 cm^−1^ also represents a strong C=O stretching bond and consisted mainly of primary alcohol. The peak value of 1020 cm^−1^ refers to a strong and broad CO-O-CO stretching bond and consisted of anhydride. The peak value of 932 cm^−1^ explains a strong C=C bending bond that consisted of the alkene compound, which is trans-disubstituted [30]. The peak value of 859 cm^−1^ explains a strong C-Cl stretching bond and it is made up of halo compounds. The peak of 764 cm^−1^ refers to a medium C=C bending bond that consisted of alkenes in tri-substituted form. The peak value of 716 cm^−1^ refers to a strong C=C bending also made up of alkenes but in a cis-disubstituted position [31,32]. The last peak value of 583 cm^−1^ for cassava nanoparticles represents a strong C-I stretching bond that consisted of halo compounds.

The analysis of PYNPs can be interpretated as follows: The peak value of 3372 cm^−1^ represents a medium N-H stretching bond that consisted of aliphatic primary amines. The value of 2936 cm^−1^ refers to a medium C-H stretching bond and indicates that the main compound was alkane. The peak of 1652 cm^−1^ represents a strong C=O stretching bond composed of lactam. The peak value of 1345 cm^−1^ refers to a medium O-H bending bond that consisted of alcohol. The peak value of 1247 cm^−1^ indicates a strong C-O stretching bond that included the alkyl-aryl-ether compound [33,34]. The value of 1161 cm^−1^ refers to a strong C-O stretching that consisted of tertiary alcohol. The peak of 1086 cm^−1^ refers to a strong C-O stretching that consisted of aliphatic ether. The peak of 1026 cm^−1^ refers to a strong S=O stretching bond that consisted of sulfoxide. The peak value of 939 cm^−1^ is related to a strong C=C bending bond that belongs to alkene compounds in the trans-disubstituted position. The value of 857 cm^−1^ is related to a strong C-Cl stretching bond which is related to a halo compound. The peak value of 770 cm^−1^ did not refer to any compound, and this is possible because not all frequencies refer to a related compound [35]. The same applies to the peak value of 719 cm^−1^. The peak of 581 cm^−1^ refers to a strong C-I stretching bond that belongs to a halo compound. Finally, the peak value of 516 cm^−1^ refers to a strong C-Br stretching bond that belongs to a halo compound [36].

### 2.4. DSC Thermogram

The basic characteristics that can been seen from Figure 8 are summarized in the following points:The glass transition region was not clear and continuous, as seen from the upper left side of the curve, and this gave the impression that these nano-polymers were more likely to have well-recognized crystalline regions during the heating process compared to the amorphous region.Due to the sensitivity of the nano-polymers that were made from the starch extracted from purple yam and cassava, the crystallinity regions could interfere with the melting regions. For this reason, there was no distinguished region specialized for the melting part; therefore, the crystallinity temperature can be considered the same as the melting temperature, which was 97.6 °C for PYNPs and 97.8 °C for CSNPs, as seen from the curve.There were high similarities between the components and structures of PYNPs and CSNPs, in such a way that the DSC thermographs were similar. The CSNPs curve was somehow higher than that of PYNPs. In this sense, the melting temperature was nearly the same for both (difference of 0.2 °C). From another aspect, choosing PYNPs for polymer flooding for the first time to improve oil recovery is not far-fetched as CSNPs have already been tested before in polymer flooding and good results for oil recovery have been obtained [7,37].

### 2.5. Rheological Properties of PYNPs and CSNPs

As observed from the viscosity measurements, viscosity increased with the increasing shear rate for both PYNPs and CSNPs polymer combinations. The measurements were conducted for a moderate shear rate to obtain the relation between the shear rate and viscosity. It was noticed that the concentration of 1.25 wt.% PYNPs in the solution that also contained HPAM (0.2 wt.%) yielded a higher viscosity, whereas at the 0.25 wt.% PYNPs concentration, the measured viscosity was the lowest until the shear rate approached 500 s^−1^. An approximately 0.75 wt.% PYNPs solution was considered the best concentration that may led to a lower viscosity value from the rheological measurement analysis. The variation of viscosity for HPAM with CSNPs was highlighted at the concentration of 1 wt.% CSNPs, which resulted in a lower viscosity at shear rate values ranging from 300 to 850 s^−1^. Therefore, for this variation, the practical concentration for the cassava nanoparticle solution with HPAM should be around 1 wt.%. The presence of PYNPs and CSNPs improved the polymer rheology in comparison to HPAM solution to a certain degree. 

Usually, examining the rheology at low shear rate values (0.1–100 s^−1^) is considered a key point as it affects enhancement of the oil recovery. Therefore, practical and real values of shear rates that should be highlighted in any study must lay within this range. In other words, a low shear rate viscosity should be always monitored. This is because pumping a too-high viscosity solution at a low shear rate away from the wellbore might cause an undesirable pump pressure increase for the injection into the reservoir. Additionally, since the goal is to maintain a stable fluid front as well as excellent flow behavior, the application of fluids that exhibit both shear-thinning and shear-thickening behavior is preferable, and this is what occurred in the current study. The inclusion of NPs has improved the viscosity and viscoelastic properties of HPAM solution [38]. The NPs/HPAM hybrid solution showed thermal stability at T = 60 °C, which is the temperature of the examined oil reservoir. The rheology test also indicated that seeding PYNPs and CSNPs facilitated the cross-links among polymer molecules and made the hybrids more elastically dominant.

### 2.6. Effect of IFT on PYNPs and CSNPs Concentration

The IFT at 60 °C for PYNPs and CSNPs with HPAM is shown in Figure 9 and Figure 10, respectively. As seen from these figures, the IFT decreased with the increasing concentration of nanoparticles in the hybrid polymer, until it reached a minimum value at a certain (critical) concentration, from which further increasing the concentration of the nanoparticles in the hybrid polymer led to an increase of the IFT value [39]. This ‘critical’ value was 0.62 wt.% for PYNPs and 0.80 wt.% for CSNPs in the polymer solution.

These values, which represent the CMC concentration for PYNPs and CSNPs, were confirmed by measuring the viscosity of the hybrid polymer solution for different concentrations of nanoparticles (0.2 to 1 wt.%) for a period of 10 days. As shown in Figure 11, the viscosity of PYNPs was approximately stable during this period when the concentration of PYNPs was 0.6 wt.%, except on the first day, where the value was somehow higher than the others. As seen in Figure 12, the viscosity of CSNPs did not change much when the concentration of CSNPs was 0.8 wt.%. This provided further confirmation about CMC values obtained earlier from IFT measurements. Therefore, these values are considered as the optimum concentrations for the two types of nanoparticles in the polymer solution. 

### 2.7. Oil Recovery from Water and Polymer Flooding

Since Windsor type I was achieved for the combination of the output liquid from polymer flooding with CSNPs, as shown in Figure 13, therefore, water flooding or water plus polymer flooding using CSNPs/HPAM was not efficient enough in extracting large amounts of oil from the core sample, in comparison to water and polymer flooding with the PYNPs/HPAM solution, as illustrated in Table 2.

The experiments indicated that polymer flooding with HPAM and PYNPs yielded the best results in oil recovery, in comparison to the other polymer combination. Despite that there was not much difference between the two, the combination that contained PYNPs was more stable as Windsor type 3 was achieved, as seen in Figure 14. Therefore, this technology improved injection process by conventional polymer flooding with HPAM through seeding 0.6 wt.% PYNPs in the last solution. The relations between oil recovery and pore volume, and that of oil recovery with time of injection, for both kinds of solutions are shown in Figure 15, Figure 16, Figure 17 and Figure 18.

The first relation demonstrated the overall oil recovery versus the quantity of solution injected, and this quantity was expressed relative to the core pore volume. The second relation was between the overall oil recovery and the time of injection. As seen from these figures, the first region was recognized for water flooding, as water was first injected into the core sample. It was found that the quantity of water injected was not exactly 1 PV: it was 1.01 PV for flooding 1 and 1.05 for flooding 2. The reason for this difference is related to the time of injection, or more accurately, the last two to three minutes of injection [40]. As the injection process was controlled to stop at a pre-set time (each 3 min), it happens that within the last 2–3 min of flooding, 1 PV of solution has already been injected. Therefore, completing the cycle of operation resulted in excess injection of the solution (water) into the core sample. The same occurred for the quantity injected by polymer flooding with nanoparticles, demonstrating 2.03 PV for HPAM/PYNPs and 2.08 PV for HPAM/CSNPs solutions.

The pressure drop profile versus the pore volume for the complete flooding operation for the injected solutions is shown in Figure 19. The pore volume resulted from dividing the injected volume of the solutions (cm^3^) by the core’s pore volume (20.64 cm^3^). When water flooding was initiated at the beginning, there was a rise in the pressure drop, followed by a decline, until it became constant after a certain time and remained stable until the end of water flooding. The reason behind the declining trend was due to the high mobility of water in comparison to that of oil. When the EOR injection was commenced, the pressure drop began to rise for both polymer solutions, but the rise was higher for the solution that contained 0.60 wt.% PYNPs. In fact, the recovery of oil improved with the increased pressure drop, and therefore, the polymer solution that contained 0.60 wt.% PYNPs was more efficient in expelling oil than the other solution. Increasing the pressure drop showed that the PYNPs solution lowered the capillary forces that hold the oil in the pore spaces [41]. Additionally, the rise in the pressure drop during CSNPs injection occurred due to the improved viscosity for this solution, in comparison with water [42,43]. A favorable trend of oil displacement occurs when:(1)$$\frac{k_{\mathrm{rw}}\mu_{w}}{k_{\mathrm{ro}}\mu_{o}}=M\leq 1$$
where M is the mobility ratio, µ_w_ is the water viscosity (mPa·s), µ_o_ is the oil viscosity (mPa·s), and k_rw_ and k_ro_ are the relative permeabilities of water and oil, respectively (mD). The use of PYNPs and CSNPs with HPAM solution delayed approaching unity, and this signified that the oil viscosity variation was less than the water viscosity variation, resulting in a favorable mobility. As PYNPs imbibed into the sandstone core in its endeavor to expel trapped oil, the pressure drop began to build up until it reached a maximum value, as shown in Figure 19. This led to the formation of O/W emulsion, as shown in Figure 14. These results agree with an early work of Pei et al. [44], which stated that O/W emulsion can enhance the sweep efficiency by blocking the channel formed by water flooding and lowering the mobility ratio. However, this was not the case in the CSNPs injection as the emulsion was not produced, as shown in Figure 13.

Among all chemical-enhanced oil recovery methods, polymer flooding with HPAM is a straightforward technique with a lengthy commercial history and proven results [45]. Despite that the increased viscosity of the water caused better mobility control between the injected water and the hydrocarbons within the reservoir, a number of field cases have been aroused over the past few years because of HPAM degradation in high-temperature and high-salinity reservoirs. Therefore, enhancing HPAM flooding with purple yam and cassava can improve the polymer properties and increase the injectant viscosity. As seen in Figure 20, the overall oil recovery for HPAM, as conducted at the Reservoir Laboratory, was 58.27%, whereas polymer flooding with HPAM and PYNPs led to an overall oil recovery of 78.46%, and that for HPAM/CSNPs was 73.91%. This means that HPAM flooding with starch nanoparticles increased the recovery ratio by 20.19% for the first polymer and by 15.64% for the second.

## 3. Materials and Methods

### 3.1. Materials

#### 3.1.1. Buff Berea Core Samples

Five Buff Berea core samples were purchased from Atama Tech Sdn. Bhd., Skudai, Johor. An additional two cores were provided by the reservoir laboratory, School of Chemical and Energy Engineering, with properties similar to those of the Buff Berea samples. These core samples were utilized in the water and polymer flooding experiments. They have physical properties similar to those of the original sandstone in the oilfield reservoir in Sumatra, Indonesia. Table 3 shows the general properties of these core samples.

#### 3.1.2. Crude Oil

Crude oil with 31.9° API was from Langgak oilfield in Sumatra, Indonesia. The viscosity of the crude oil was around 43.668 cp, and at a normal temperature (~25 °C), the oil phase structure was in a solid state. Generally, the pour point and wax content for most Indonesian crude oils lay between 35 °C and 40 °C and 20% and 25%, respectively. Dealing with such oils requires special equipment in order to keep them in a liquid state. To overcome this problem, the oil was treated with a chemical solution called Fsol, at a ratio of 1:1. This chemical solution was purchased from an innovative company called Innochems Technologies Sdn. Bhd., located in Johor, Malaysia.

The company did not reveal the procedure used to manufacture Fsol, but this solution has the ability to reduce the viscosity of oil and keep it in a liquid state at normal temperatures, without changing its main properties. This can be achieved through mixing the crude oil with this solution in the aforementioned ratio and then stirring the diluted oil for around 10 min using a magnetic stirrer. In order to ensure the homogeneous composition of this oil after it was mixed with Fsol, it was placed in an oven at 60 °C for around 30 min before using it for injecting.

#### 3.1.3. Partially Hydrolyzed Polyacrylamide

The partially hydrolyzed polyacrylamide (HPAM) 0.5% (*w*/*v*) aqueous solution, brand R&M, was purchased from Tricell Bioscience Resources Co., located at Taman Universiti, Johor, Malaysia.

#### 3.1.4. Acetic Acid (CH_3_COOH)

Glacial acetic acid (CH_3_COOH) with a purity of 99% (*w*/*w*) was supplied by QREC (Asia) Sdn. Bhd., Selangor, Malaysia. 

#### 3.1.5. Purple Yam Tubers

A total of 18 kg of purple yam tubers was purchased from a local market. It is known scientifically as “Dioscorea Alata”, and other names are grater yam and water yam. It is one of a variety of species of yam that were domesticated and cultivated within Southeast Asia and New Guinea for their starchy tubers.

#### 3.1.6. Native Cassava Starch

A total of 1 kg of native cassava starch was purchased from a local market in Johor. Cassava is a versatile root vegetable that is widely consumed in several parts of the world, and it is also what tapioca starch is made from. Cassava starch is a white powder made from tapioca that has been dehydrated and dried after being extracted.

### 3.2. Methods

#### 3.2.1. Extraction of Purple Yam and Cassava Starch

Here, the 18 kg of tubers of purple yam was peeled, washed, and crushed to fine particles using a grinder machine. Before the grinding process, a small amount of water was added to the container of the grinding machine to ensure smooth cutting. The obtained thick solution from the grinding process was poured into a large-size vessel, passing through a 140 μm-mesh size sieve, after which it was left for precipitation to take effect after around 8 h. The fiber retained on the sieve was washed with water so that more solution containing starch could be retained on the vessel. After that, the collected starch was washed with pure water to remove any available fiber. The produced purple yam starch was dried in an oven at 45 °C for about five hours to eliminate the moisture content, and then it was sun-dried for a certain time to ensure complete drying. The sun-drying process is also useful for the bleaching of starch and for reducing the cyanide content [7].

The dry weight of the purple yam starch produced through this operation was obtained by using an electronic balance Shimadzu AY220. The produced starch was directed to pass through a 72 μm-mesh size (British standard) by using a mechanical sieve to ensure that only small particle sizes of starch granules were collected. The resulting fine starch was stored in an airtight container for analysis, and further modified to crystalline starch nanoparticles through acid hydrolysis, which was assisted by ultrasonication. A sketch for this process is shown in Figure 21.

#### 3.2.2. Synthesis of Purple Yam and Cassava Nanoparticles

A total of 1 kg of native cassava starch was purchased from a local market and was ready to process to a nanomaterial. Here, 40 g of purple yam and cassava starch were dissolved separately in 250 mL of acetic acid solution at a concentration of 2.5 mol/l in an Erlenmeyer flask. Then, the resulting solution was continuously stirred at a constant speed using a magnetic stirrer at 45 °C for 5 days. According to Table 4, the processability range was between two limits: maximum and minimum. The selected concentration for acetic acid, the temperature, and the time (in days) were selected between these two limits for better nanoparticle formulation. After that, the solution was put in an ultrasonic instrument, CREST LL TRANSONICS, with a frequency of 40 kHz and a power output of 500 W for one hour. The use of ultrasonication with high intensity can help in reducing the molar mass of the solute and preventing nanoparticles’ aggregation [46].

The resulting solution was centrifuged for 20 min at a constant rate of 4000 rpm using the ROTOFIX 32A instrument, and the supernatant fluid was removed. The produced nanoparticles were washed twice in order to remove the remaining acetic acid. Produced nanoparticles were placed in an oven for 24 h at 40 °C for the drying process. Figure 22 represents the aforementioned mechanism of production. Three independent parameters could affect the production of nanoparticles: acid concentration (mol/l), temperature (°C), and time (days), whereas the output dependent responses were the yield (%) and the particle size (nm) of the produced nanoparticles. The accessibility range for these independent parameters was determined according to previous studies [38,47,48], as shown in Table 4.

#### 3.2.3. Particle Size Distribution (PSD)

The yield of the produced nanoparticles from each kind can be calculated according to the following equation [49]:Yield (%) = (W_CSNP_/W_NS_) × 100%(2)
where W_CSNP_ represents the final weight of the produced nanoparticles after the completion of the drying process, and W_NS_ is the initial weight of the native starch for cassava and purple yam, which was 40 g.

Particle size distribution yields important information about the size of particles and their geometrical shapes (spherical, hexagonal, or longitudinal). This valuation was conducted at the University Industry Research Laboratory (URIL UTM). Laboratory evaluations for the particle size were performed based on three major variables: intensity, volume, and the number percent of these nanoparticles.

#### 3.2.4. Surface Charge for Nanoparticles

The stability of particles in a certain solution can be quantified by measuring the Zeta potential for the particles. By using electrophoretic light scattering, the velocity of the particles can be determined by measuring the frequency shift of the light scattered by the motion of the particles. It was agreed that if the absolute Zeta potential values were over 60 mV, then the particles had excellent stability, whereas if the values were above 30 mV, the particles were physically stable, accordingly. Furthermore, if the values were lower than 5 mV, this is an indication for the agglomeration of particles [49,50], as expressed in Table 5 [51].

#### 3.2.5. Polymer Rheology Analysis

It is important to characterize the polymer rheology to obtain a clear picture about its suitability for the injection process. This was carried out through the use of the Brookfield RST rheometer, from which the viscosity of the HPAM polymer, purple yam, and cassava nanoparticles were measured and plotted versus the shear rate. These solutions were heated to the reservoir temperature 60 °C.

Approximately 68.5 mL of polymer solutions consisted of HPAM at a concentration of 2000 ppm, and then HPAM (2000 ppm) with PYNPs and with CSNPs were placed separately inside the cylindrical tube of the rheometer. The concentration of the purple yam/cassava nanoparticles varied from 0.25 to 1.25 wt.%. A number of measurements were obtained for a certain range of the shear rate (300–1000 s^−1^). 

#### 3.2.6. Optimum Concentration of Nanoparticles

In order to find the critical micelle concentration (CMC) for the produced PYNPs and CSNPs, different concentrations (0.2, 0.4, 0.6, 0.8, and 1 wt.%) were prepared by dissolving these particles in a brine solution (100 ppm). Then, 2000 ppm of HPAM was added to the nanoparticle solution. The hybrid polymer from each type was placed in contact with paraffin oil at 60 °C. Using a paraffin oil yielded an acceptable approximation to the actual oil conditions in the reservoir oilfield [52]. The interfacial tension between the paraffin oil and the hybrid polymers was calculated using the KRUSS EasyDyne tensiometer (K20).

#### 3.2.7. IFT Measurements

The interfacial tension (IFT) for different concentrations (0.2, 0.4, 0.6, 0.8, and 1 wt.%) of nanoparticles in the hybrid solution was measured in order to find the CMC, which represents the optimum concentrations for PYNPs and CSNPs. The concentration of HPAM solution was held constant at 2000 ppm. The expected relation between the IFT and the concentration is that the IFT should decrease with the increasing solution concentration until it reaches a minimum value. The corresponding value of the concentration at this minimum value for the IFT was the CMC for that nanoparticle solution.

#### 3.2.8. Flooding Experiments

Flooding was initiated by inserting two hybrid polymers into the core flooding instrument. The first nano-polymer consisted of 2000 ppm HPAM plus 0.60 wt.% PYNPs. The second polymer solution consisted of 2000 ppm HPAM and 0.80 wt.% CSNPs. Three Buff Berea core samples were placed inside the saturation vessel and allowed to be vacuumed. Then, the vessel was connected to a vacuum pump for three hours to withdraw the air from inside the core samples. Brine was prepared at 100 ppm by dissolving 10 g of NaCl in 1000 mL of of distilled water. Brine was carefully injected inside the saturation vessel by means of the same vacuum pump.

When the brine solution fully filled the free space inside the saturation vessel, the excess brine began to discharge out from the vessel and collect inside a conical flask. Then, the vacuum pump was turned off and the saturation vessel was connected to a Teledyne pump. The pressure inside the vessel was gradually increased by the injection of 10 cm^3^/min of brine through the Teledyne pump. When the pressure inside the saturation vessel reached around 2000 psi, the brine injection from the accumulator was stopped. The vessel remained under this pressure for two to three days in order to fully saturate the core samples with the brine. 

After the cores were completely saturated with brine, they were ready to be saturated with the crude oil. The brine was removed from the accumulator and the saturation vessel was opened, and two core samples were removed and placed in a 1000 mL beaker containing brine to maintain their saturation. After that, the remaining core sample was inserted into the confining vessel and the crude oil that was previously mixed with Fsol was injected. Nitrogen gas was pumped into the confining vessel to assist in spreading the oil inside the sandstone pores. The Teledyne pump was operated at a volumetric flow rate of 8 cm^3^ per minute to inject the oil inside the confining vessel. The output from the injection was settled in a 50 mL cylinder. After the oil saturated the core sample, it began to push on the water (brine) from inside the core, allowing it to exit and accumulate in the cylinder. This process was continued until excess oil (just few drops) was discharged [53]. At this point, the oil injection process was stopped, and the volume of water produced during the oil injection was equal to the original oil in place (OOIP). The basic principle behind this is related to the law of the conservation of mass.

After knowing the quantity of the original oil in place (OOIP), the volume of water still inside the sandstone core after the oil injection/saturation could be estimated. That is the irreversible water content that could not be retained after the oil injection. Therefore, the initial oil saturation was always less than 100% under these conditions. After the saturation of the core sample with the crude oil, water flooding was initiated by an injection of 5 cm^3^/minute of brine into the core sample. The output from the water flooding was accumulated in the 50 mL cylinder, and its quantity was estimated every 3 min. The water flooding was then resumed until one pore volume (1 PV) of water was injected [54].

The polymer flooding was initiated after water flooding, and it was related to the injection of the two aforementioned HPAM/nano polymers. Then, 3 to 3.5 cm^3^/min of nano-polymer was injected, and the output to the collecting cylinder was checked every three minutes, as in water flooding. In order to maintain the temperature for water and polymer flooding at around 60 °C, the confining vessel was placed inside an electrical oven after setting its temperature to 60 °C. The temperature inside the oven was verified to be around 60 °C by measuring the temperature through a thermocouple. The polymer flooding was continued until 2 PV [55] of nano-polymer was injected inside the core sample. The oil recovery percent for the recovery process (RF) can be calculated from the following equation [56]:RF = (volume of oil produced at cylinder/OOIP) × 100%(3)

## 4. Conclusions

In this study, new natural materials were involved in polymer flooding to improve the recovery of oil in comparison to the conventional polymer flooding that employed HPAM. These natural materials were obtained from purple yam and cassava starches and their sizes were decreased to the nano-scale to improve their properties. The average particle size for PYNPs was 363.12 nm and that for CSNPs was 52.92 nm based on the intensity percent measurements.

According to Zeta potential distribution, PYNPs were more stable in comparison to CSNPs as the mean Zeta potential was −36.3 mv for PYNPs and −10.7 mv for CSNPs, respectively. The yield of the produced PYNPs was 85%, and that for CSNPs was 90.53%. From the IFT and polymer rheology measurements, it was confirmed that the optimum concentration of PYNPs was 0.60 wt.%, and that of CSNPs was 0.80 wt.%. 

It can be concluded that produced nanoparticles from each kind has improved the oil recovery by involving them in polymer flooding with HPAM after water flooding. Additionally, percentage of oil recovered from the PYNPs solution was higher than that obtained from CSNPs (33.46% for PYNPs and 31.3% for CSNPs). This was also confirmed from phase equilibria, as Windsor type 3 was achieved using the HPAM/PYNPs solution. Windsor type 3 is preferable in the oil industry, and it refers to an efficient oil recovery [42]. Thus, the overall oil recovery based on HPAM/PYNPs solution has increased to 78.46%, and that for the other polymer combination increased to 73.91%, whereas polymer flooding with HPAM alone led to an overall oil recovery of around 58.27%.

## Figures and Tables

**Figure 1 molecules-28-04614-f001:**
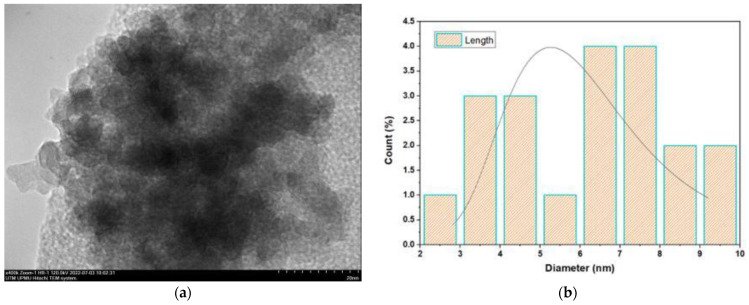
Sample of PYNPs: (**a**) TEM spectra and (**b**) particle size distribution.

**Figure 2 molecules-28-04614-f002:**
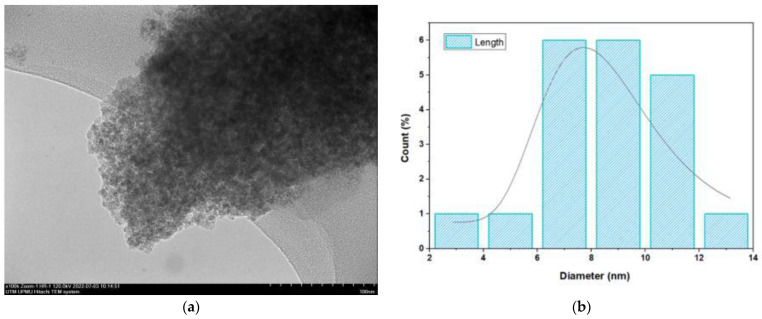
Sample of CSNPs: (**a**) TEM spectra and (**b**) particle size distribution.

**Figure 3 molecules-28-04614-f003:**
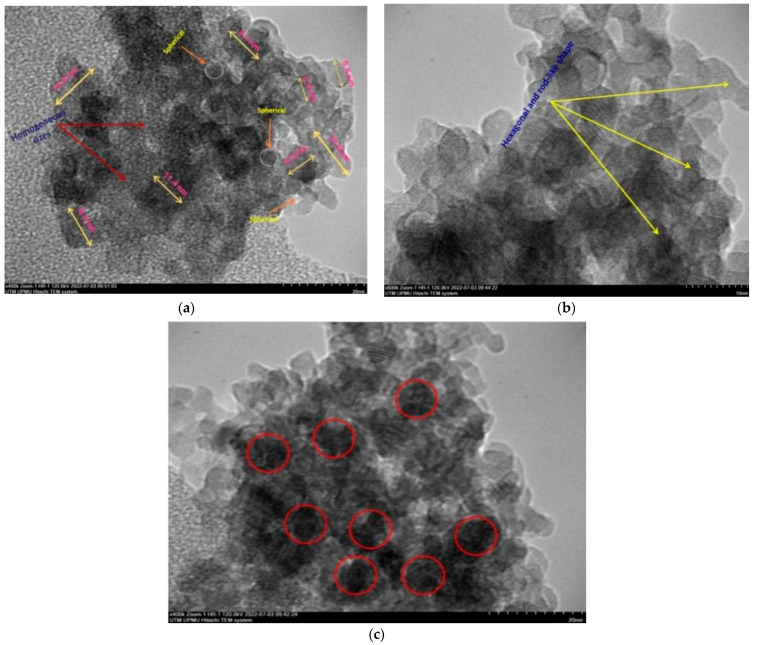
TEM images for PYNPs: (**a**) homogenous-sized particles, (**b**) hexagonal and rod-like shape, and (**c**) nucleation due to cavitation bubbles.

**Figure 4 molecules-28-04614-f004:**
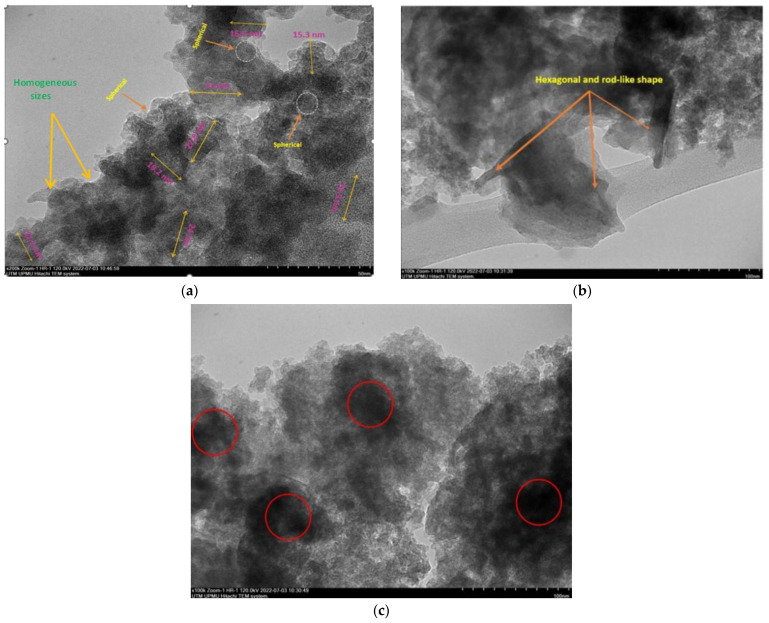
TEM images for CSNPs: (**a**) homogenous-sized particles, (**b**) hexagonal and rod-like shape, and (**c**) nucleation due to cavitation bubbles.

**Figure 5 molecules-28-04614-f005:**
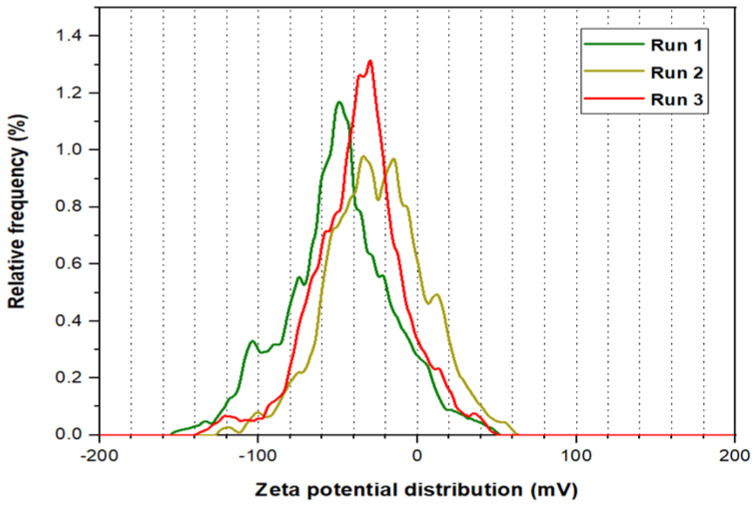
Zeta potential distribution for PYNPs.

**Figure 6 molecules-28-04614-f006:**
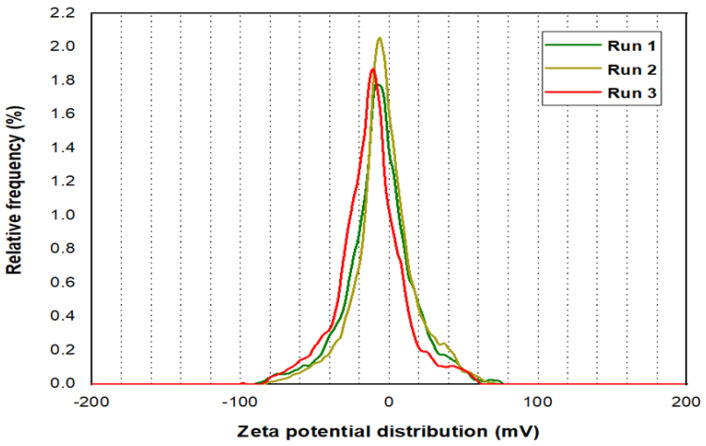
Zeta potential distribution for CSNPs.

**Figure 7 molecules-28-04614-f007:**
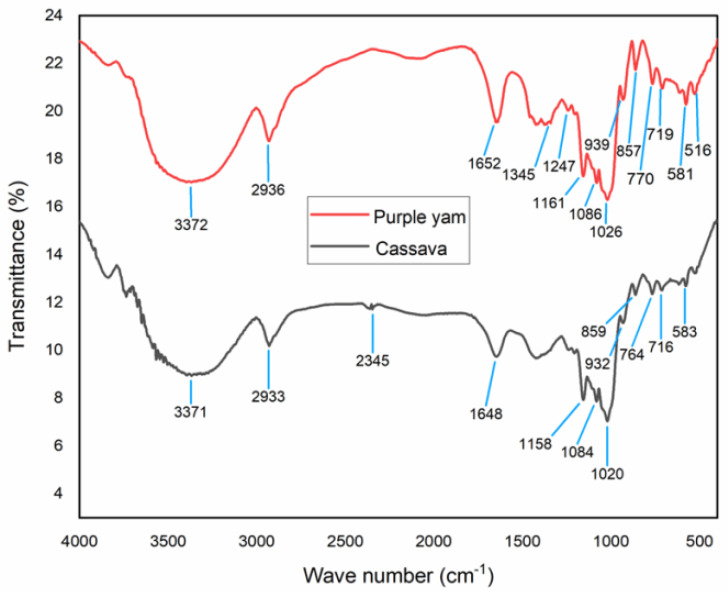
Relation between transmittance and wave number for PYNPs and CSNPs.

**Figure 8 molecules-28-04614-f008:**
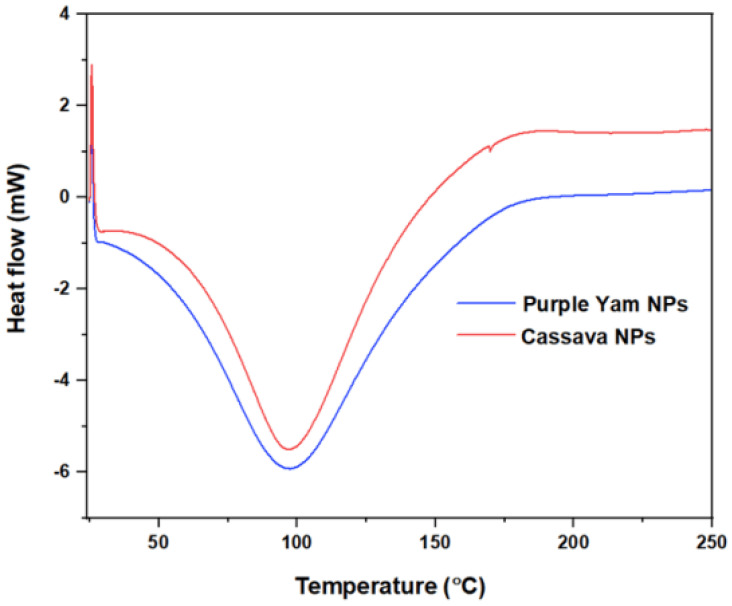
DSC thermograph for PYNPs and CSNPs.

**Figure 9 molecules-28-04614-f009:**
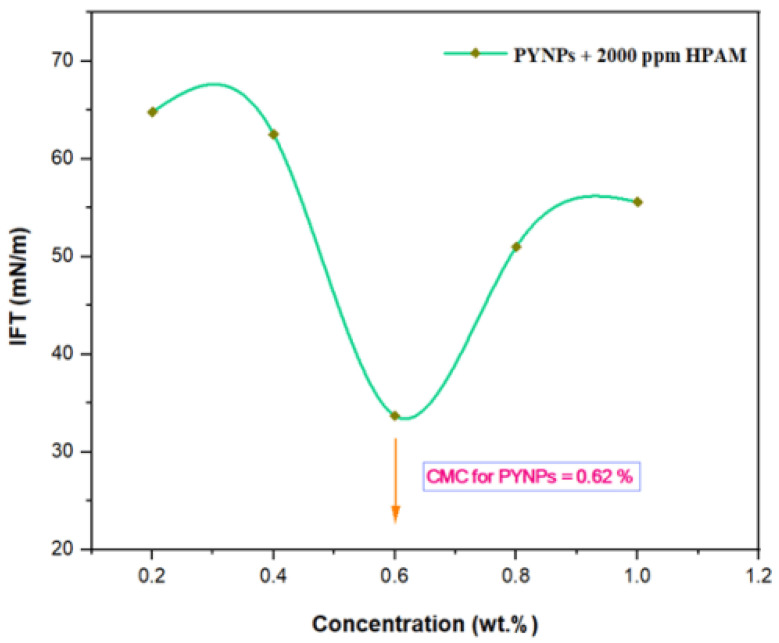
Relation between the IFT and the concentration for the HPAM/PYNPs solution at 60 °C.

**Figure 10 molecules-28-04614-f010:**
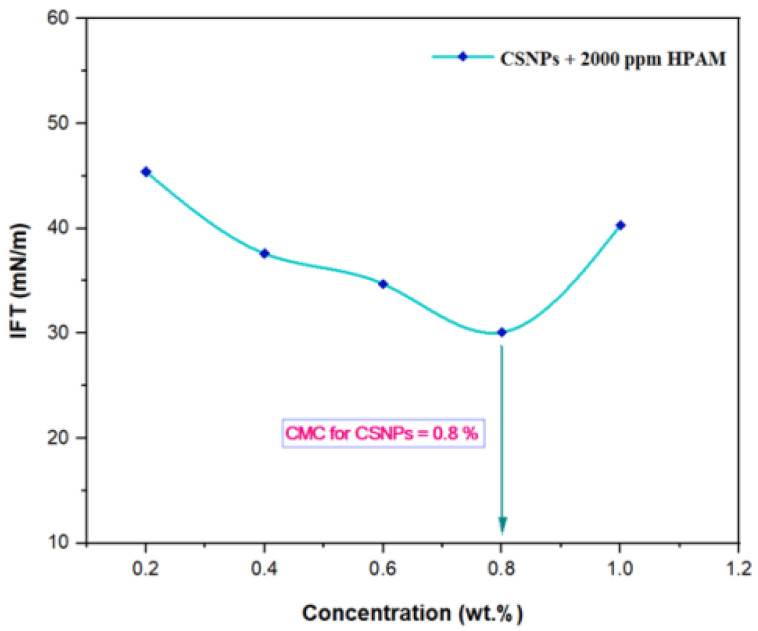
Relation between the IFT and the concentration for the HPAM/CSNPs solution at 60 °C.

**Figure 11 molecules-28-04614-f011:**
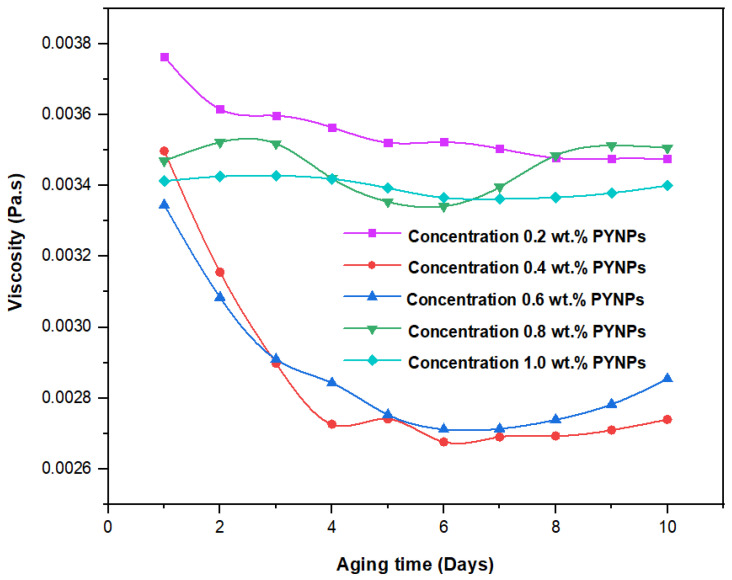
Relation between the viscosity of the HPAM/PYNPs solution and the aging time (in days) for five concentrations of PYNPs.

**Figure 12 molecules-28-04614-f012:**
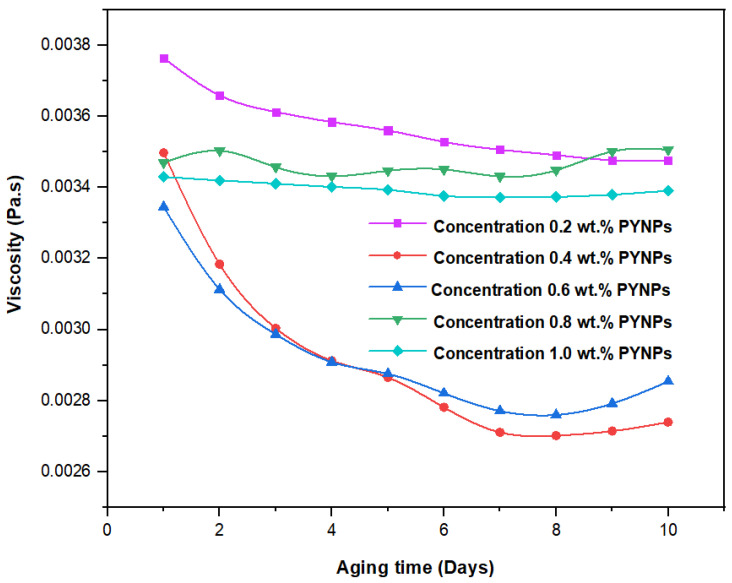
Relation between the viscosity of the HPAM/CSNPs solution and the aging time (in days) for five concentrations of CSNPs.

**Figure 13 molecules-28-04614-f013:**
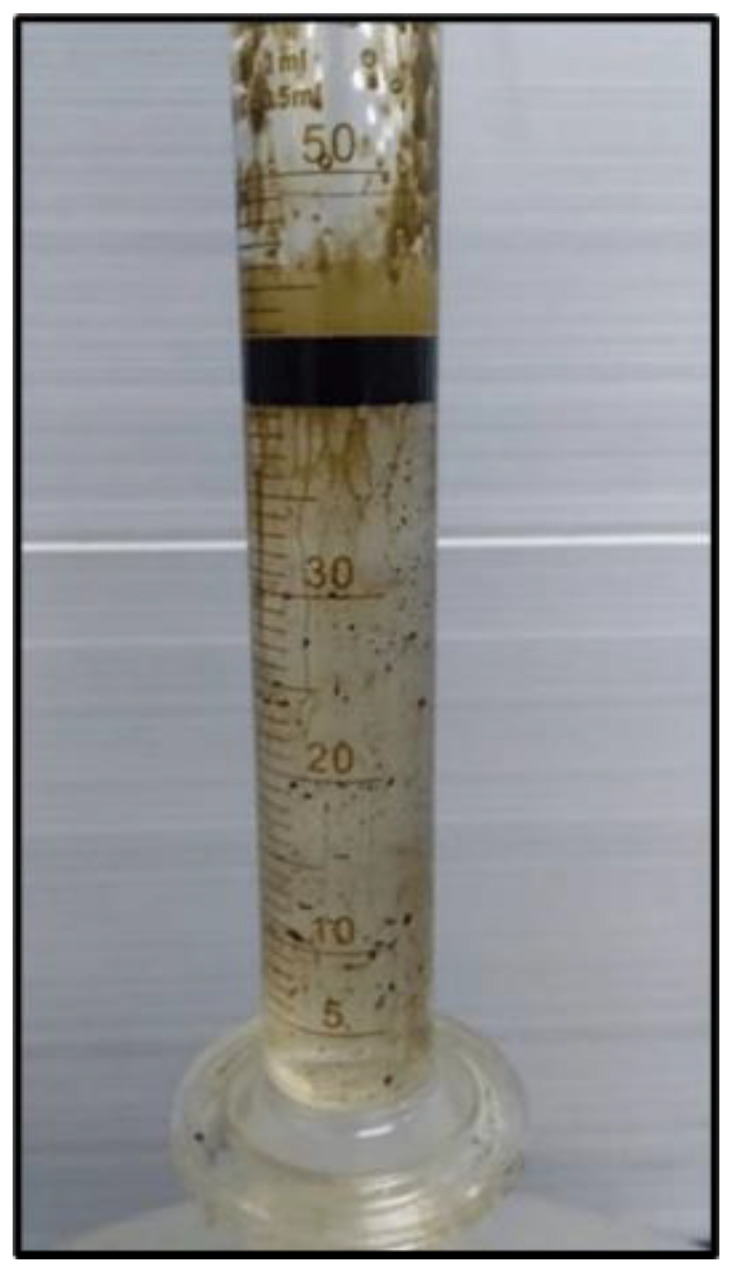
Oil extracted after water and polymer flooding at 60 °C. The polymer solution consisted of 2000 ppm of HPAM and 0.8 wt.% CSNPs.

**Figure 14 molecules-28-04614-f014:**
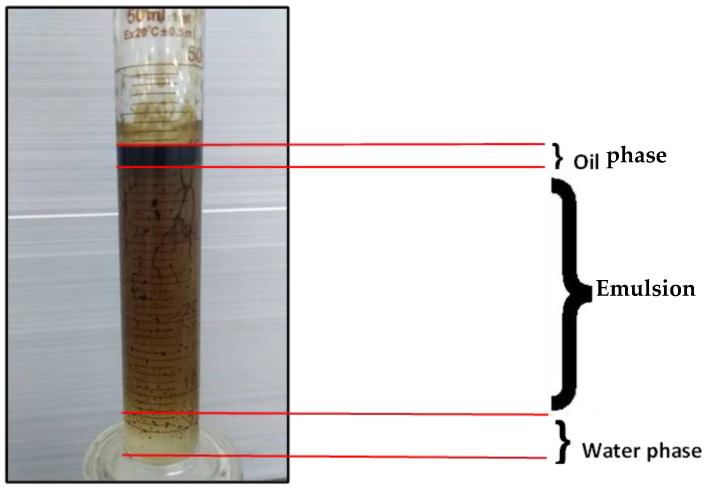
Oil extracted after water and polymer flooding at 60 °C. The polymer solution consisted of 2000 ppm of HPAM and 0.6 wt.% PYNPs.

**Figure 15 molecules-28-04614-f015:**
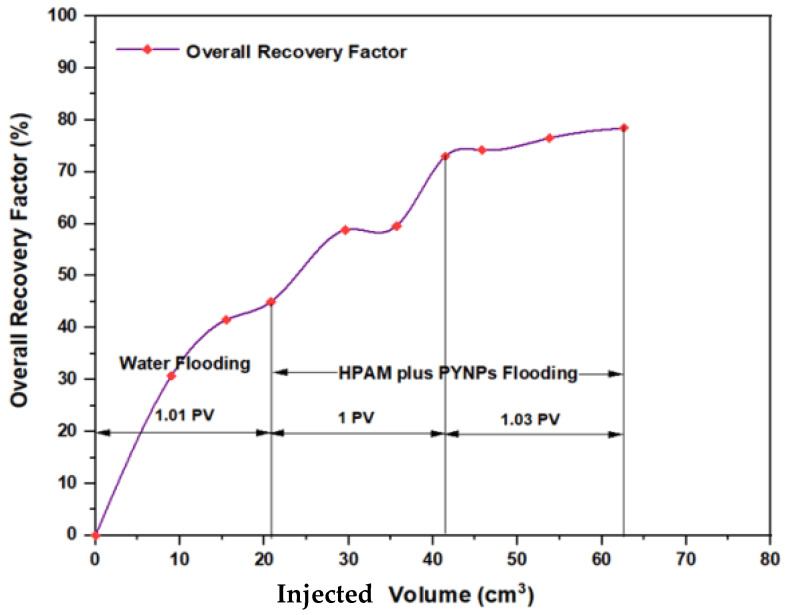
Overall oil recovery versus injected volume from water and polymer flooding at 60 °C. The polymer solution consisted of 2000 ppm of HPAM and 0.60 wt.% PYNPs.

**Figure 16 molecules-28-04614-f016:**
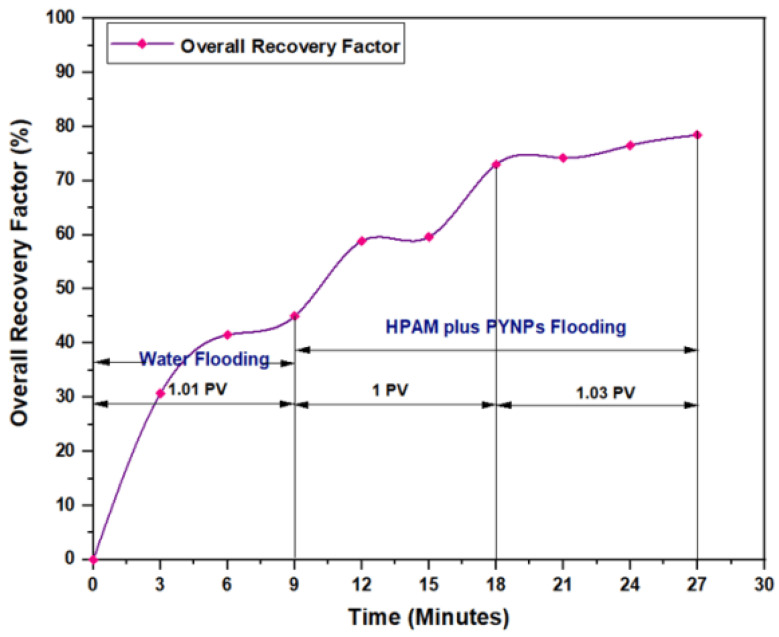
Overall oil recovery versus time of injection from water and polymer flooding at 60 °C. The polymer solution consisted of 2000 ppm of HPAM and 0.60 wt.% PYNPs.

**Figure 17 molecules-28-04614-f017:**
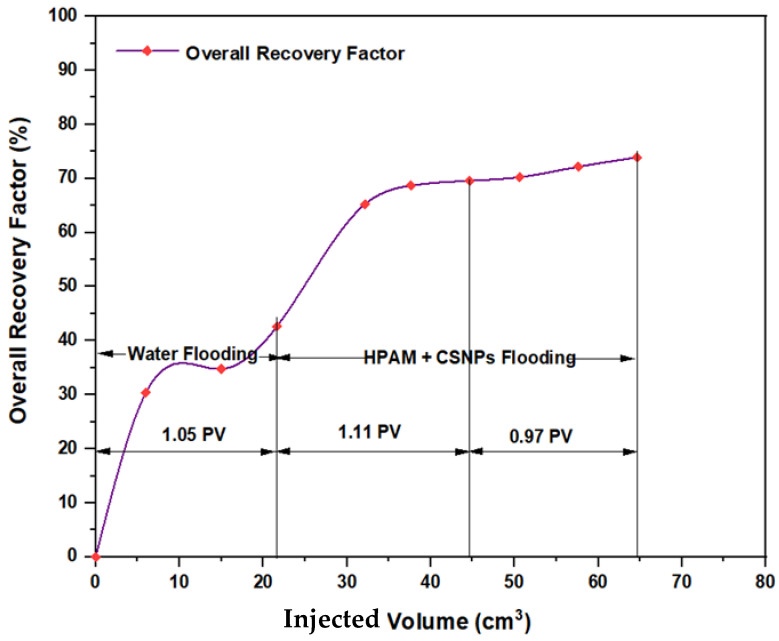
Overall oil recovery versus injected volume from water and polymer flooding at 60 °C. The polymer solution consisted of 2000 ppm of HPAM and 0.80 wt.% CSNPs.

**Figure 18 molecules-28-04614-f018:**
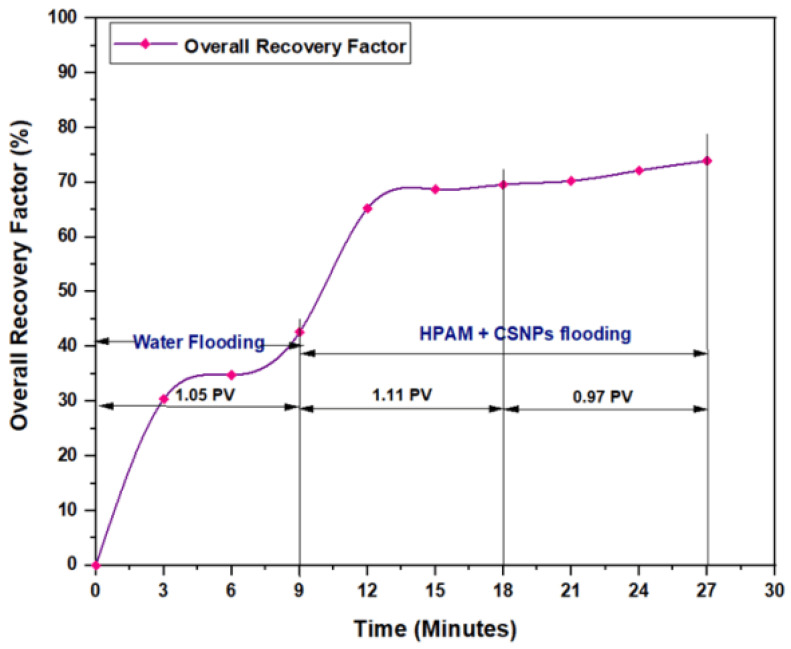
Overall oil recovery versus time of injection from water and polymer flooding at 60 °C. The polymer solution consisted of 2000 ppm of HPAM and 0.80 wt.% CSNPs.

**Figure 19 molecules-28-04614-f019:**
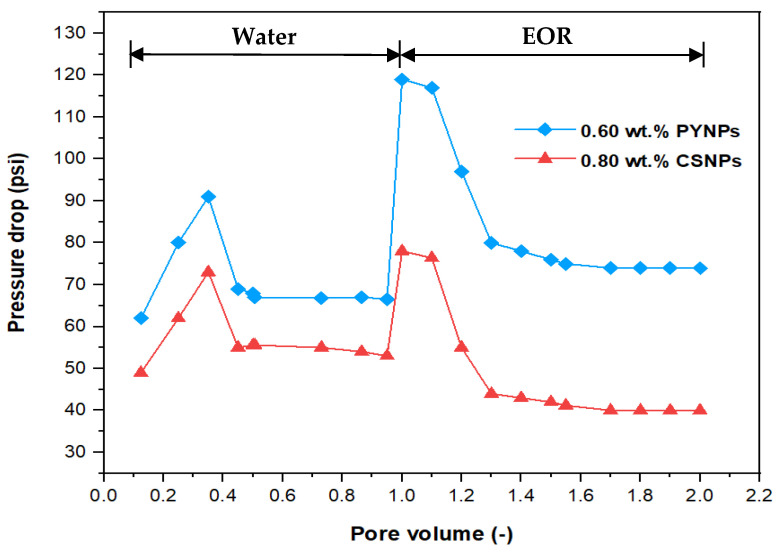
Pressure drop versus pore volume for the water and polymer injections at 60 °C. Polymer solutions consisted of 2000 ppm of HPAM with 0.80 wt.% CSNPs and with 0.60 wt.% PYNPs.

**Figure 20 molecules-28-04614-f020:**
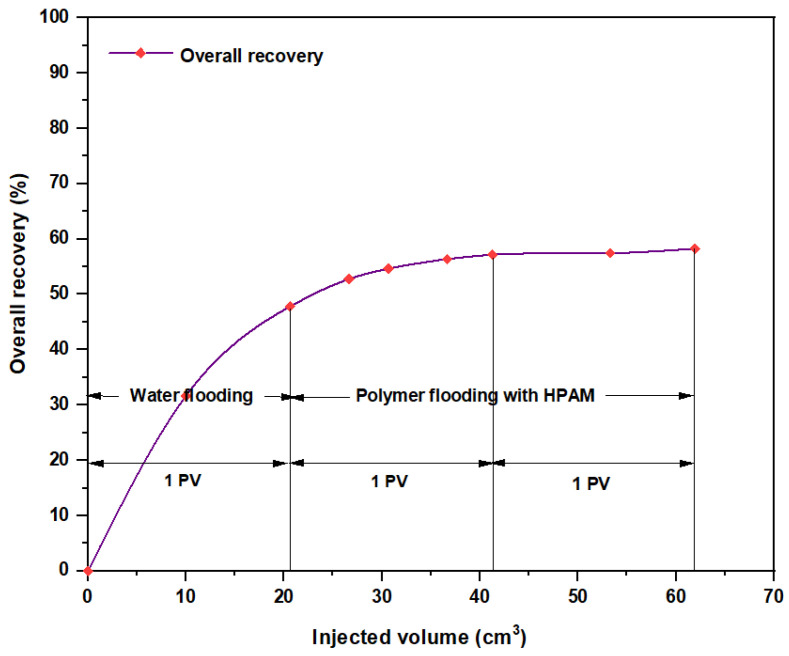
Overall oil recovery versus injected volume from water and polymer flooding at 60 °C. The polymer solution consisted of 2000 ppm of HPAM.

**Figure 21 molecules-28-04614-f021:**
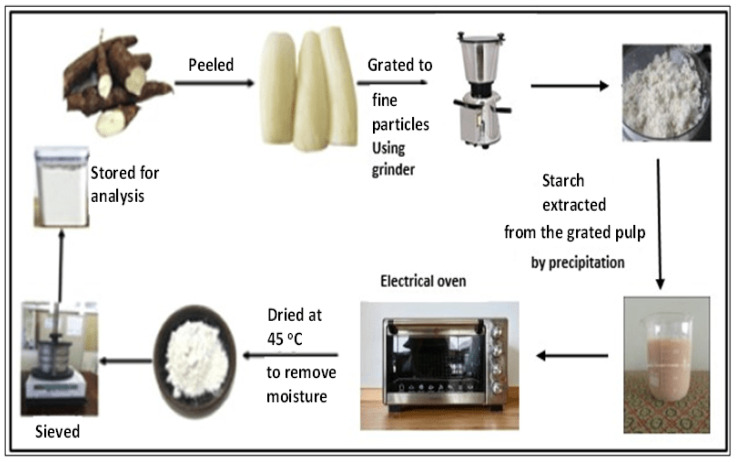
Purple yam starch preparation cycle from purple yam tubers.

**Figure 22 molecules-28-04614-f022:**
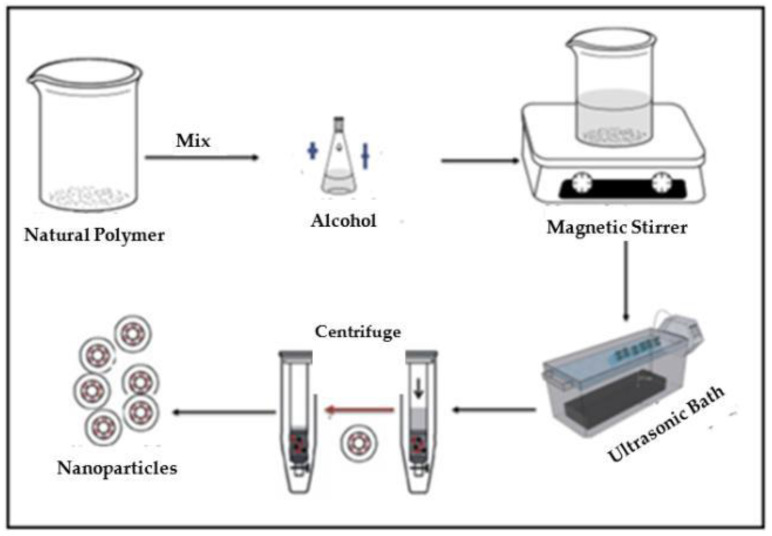
Synthesis process of starch nanoparticles from PYS and CAS.

**Table 1 molecules-28-04614-t001:** Properties of the produced nanoparticles.

NPs Type	Mean Particle Size (nm)	Mean PDI	Mean Zeta Potential (mv)	Stability Status
PYNPs	363.12	0.937	−36.3	Moderate (more stable particles)
CASNPs	52.92	0.916	−10.7	Incipient (less stable particles)

**Table 2 molecules-28-04614-t002:** Oil recovery results from water and polymer flooding at 60 °C.

Flooding 1	RF%	Flooding 2	RF%
Water flooding ^1^	45	Water flooding ^2^	42.61
Polymer flooding using HPAM/PYNPs	33.46	Polymer flooding using HPAM/CSNPs	31.3
Overall recovery (water + polymer) flooding	78.46	Overall recovery (water + polymer) flooding	73.91

^1^ Before implementing polymer flooding with HPAM/PYNPs. ^2^ Before implementing polymer flooding with HPAM/CSNPs.

**Table 3 molecules-28-04614-t003:** Buff Berea core sample properties.

**Product ID**	SS-104
**Formation**	Upper Devonian
**Permeability**	150–350 mD KCL 400–500 mD N_2_
**Porosity**	20–22%
**UCS**	3800–4500 psi
**Homogeneous**	YES
**Perm by**	KCL/N_2_

**Table 4 molecules-28-04614-t004:** Independent variables and their limits for optimum production of nanoparticles.

Acid Hydrolysis Parameters (Independent Variables)	Processability Ranges	
	Minimum	Maximum
Acid concentration, mol/L	2.2	3.6
Temperature, °C	40	60
Time, days	3	7

**Table 5 molecules-28-04614-t005:** Stability behavior of colloids according to the value of Zeta potential [51].

Magnitude of Zeta Potential (mV)	Stability Behavior
0 to 5	Rapid coagulation of flocculation
10 to 30	Incipient instability
30 to 40	Moderate stability
40 to 60	Good stability
>61	Excellent stability

## Data Availability

Not applicable.

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
