# Peer review of "Enhancing Oil Recovery by Polymeric Flooding with Purple Yam and Cassava Nanoparticles"

_molecules, 2023, doi:10.3390/molecules28124614_

Round 1
Reviewer 1 Report
In the paper under review two natural polymers prepared in the form of nano-particles are proposed for use in the compositions for enhancing oil recovery,.
These materials are carefully and completely studied by all standard structure methods. The concentrations of both polymers are chosen to be less than 1%.. In the final stage of the study, the flooding experiments have been carried out. The comparison of the new products with solutions of HPAM shows that their viscosities are close.
The experimental results are presented in a rather elegant and easily accepted form.
In general, quite interesting and accurately executed work is presented, which shows the promise of using new natural polymers for enhanced oil recovery.
However I have the following comments, which should be taken into account at the final prelaring the paper for publication.
1. A very important 9and necessary) requirement for liquids proposed for EOR is the existence of the yield stress for carrying the propant. For this purpose, the relevant experimental data related to the region of low shear stresses should be presented. In this aspect, Figs 11 and 12 are unsatisfactory since they do not prove anything.
2. I strongly against presenting experimental data by points joint by segments of broken lines instead of the use of averaging smooth curves (Figs 11, 12, 15, 16). This method (although rather widely used) is principally wrong by its physical sense because all real physical dependencies are smooth., and this hides of presentation hides possible experimental errors and does nor says anything about confidence limits of the experiments.
3. Conclusion is too long, Indeed, Conclusions must not repeat the content of the experiments but only which conclusions can be made on their base. So, it is necessary to shorten and remake this part.
Author Response
Dear Milica Stanimirovic,
Greetings to you.
Thank you very much for your significant efforts and follow up. We have finished the required changes to our research work based on the reviewers' comments. In the list below our point to point response for the valuable notes suggested by the respected reviewers. Thanks for your valuable efforts and please accept our best regards and appreciation.
Authors
Reviewer #1:
Dear Reviewer #1, Thank you for your wonderful comments which gave our work more strength. We did our best and made the necessary changes based on your valuable comments. The changes are in tracked mode in the revised MS word manuscript.
Thank you in advance for your valuable efforts with us.
Best regards,
Authors
Reviewer #1 comments:-
- A very important and necessary) requirement for liquids proposed for EOR is the existence of the yield stress for carrying the propant. For this purpose, the relevant experimental data related to the region of low shear stresses should be presented. In this aspect, Figs 11 and 12 are unsatisfactory since they do not prove anything.
Answer: Thank you for this valuable notice. We revised the section related to the rheological properties of PYNPs and CSNPs and excluded the discussion about the higher shear stresses and focused more in the results obtained in the region of low to moderate shear stresses. In this regard, Figs 11 and 12 have been deleted from this paragraph.
2. I strongly against presenting experimental data by points joint by segments of broken lines instead of the use of averaging smooth curves (Figs 11, 12, 15, 16). This method (although rather widely used) is principally wrong by its physical sense because all real physical dependencies are smooth., and this hides of presentation hides possible experimental errors and does nor says anything about confidence limits of the experiments.
Answer: Thank you for this valuable information. We have already deleted Figs 11 and 12 as illustrated above and redrew Figs 15 and 16 using another drawing pattern and also decreased the limits of Y scale so that it may show any possible experimental errors (if found). In doing so, the previous Figs 15 and 16 have been deleted.
3. Conclusion is too long, Indeed, Conclusions must not repeat the content of the experiments but only which conclusions can be made on their base. So, it is necessary to shorten and remake this part.
Answer: Thank you very much for this valuable info. We have already reduced the information available in the conclusion and focused only on the key and basic findings.
Reviewer 2 Report
This paper addresses a relevant topic, namely, Enhancing Oil Recovery by Polymeric Flooding with Purple Yam and Cassava Nanoparticles. The process entails synthetizing, characterizing and investigating performance of nano polymeric materials to enhance oil recovery from heavy oil reservoir. Nano polymeric particles were extracted from purple yam and cassava starches as raw materials. FTIR, TEM, DSC and ZETA potential were used to characterize the nano polymeric particles. Performance testing was done with a core-flood rig after mixing with common synthetic polymer HPAM. The paper is well organized and interesting to read. The abstract, however, is not impactful; it does not describe the study's purpose or significance. The reasons for integrating HPAM with nano polymers are unclear. The Results section should include the strengths and limitations of the study, as well as including other publications that support or contradict the authors' findings to convince readers about the importance of the findings. Below are some suggestions and comments.
1- Line #33 please write was instead of is.
2- Line # 59, “high number of works that are devoted to develop the bio-composites by blending such starch nanoparticles into the biopolymeric matrices”. Please cite those works and explain why you chose to integrate the starch nanoparticle with synthetic HPAM and not with biopolymer?
3- Line #87-88, 91and 100, please write “at the same time” instead of “in the same time” “improvement” instead of “improve” , ‘kinds of’ instead of ‘kinds on’ ‘as a function of’ instead of ‘as function of’
4- Line # 104 “has resulted in obtaining the highest grafting percentage and water solubility, which were 1565.53 and 96.06%, respectively. Please revise the sentence and the numbers.
5- Line # 113, I think polymer concentration should be chosen based on oil viscosity to control mobility and improve sweep efficiency, so what is the reason behind selecting 2000 ppm HPAM polymer?
6- Line # 115, “optimum concentrations on the two types of nano polymer were selected based on contact angle” please explain? Refer to comment #5.
7- Line # 145, please write Replace “composition of this oil after it was mixed with Fsol” instead of “composition for this oil after it mixed with Fsol”
8- Line # I think oil viscosity at reservoir condition is important when designing a polymer flooding rather than using viscosity at 25 °C as reported in the methodology section.
9- Line # 136, since oil was solid at normal temperature, I believe it’s a waxy crude so wax content and pour point should be added to oil properties for better understanding to readers. Also, the impact of additive “Fsol” on oil viscosity should be reported.
10- It would be useful to include the composition of crude oil, core mineralology, and make-up brine for polymer solution since polymer behavior differs when it interacts with different types of rock and fluids.
11- Line # 148 and 152, “Partially hydrolyzed polyacrylamide (HPAM) 0.5% aqueous solution”, 0.5% is it w/w, w/v? Same for “Glacial acetic acid (CH3COOH) with purity of 99%”.
12- Line # 153 “All chemicals” Which chemicals are referred to by "All chemicals" is not clear.
13- Line # 172, 174, Do you mean fiber or fibber ?
14- Line # 199, 229 I think that Figure 1 and 2 are unclear, need to improve resolutions
15- Line #203, The basis of selecting these values 2.5 mol/l acid concentration, 45°C and 5 days is unclear in the article, please elaborate.
16- Line # 239 “geometrical shapes”! please explain
17- Line # 252 Please add citation to title of Table 3.
18- Line # 260 I think there is inconsistency of using 5000 ppm HPAM for viscosity measurement, 2000 ppm HPAM for coreflood. Please explain this inconsistency in the concentrations.
19- Line # 263: In situ shear rate for fluid flow inside reservoir far from wellbore around ~ 7-10 S-1, it is imperative to include low shear rates. As polymer solution behaviour, pertaining to the shear stress and shear rate, will help in determining the behaviour of the displacing fluids for EOR process, whether it is Newtonian or non-Newtonian.
20- Line # 246: CMC commonly used for surfactant not polymer that yields utra-low IFT. or polymers and nanoparticles, it is recommended to choose their concentrations based on their viscosity, injectivity, adsorption, and costs.
21- Line # 303, 315, 320, I think there is also inconsistency in flow rates used for oil flood (8 cm3 per minute), brine flood (5 cm3/minute) and polymer (3 to 3.5 cm3/minute). Please explain
22- Line #333, 90.53 and 85%, w/w.% or v/v%?
23- Line # 340 “The average particle size for PYNPs and CSNPs based on intensity percent (which is useful in detecting small amounts of aggregation) was 363.12 and 52.92 nm, respectively” this sentence contradicts with zeta potential results -36.3 and -10.7 for PYNPs and CSNPs, please explain.
24- Line #571, “From these figures, the viscosity increased with increasing shear rate for both PYNPs and CSNPs” it seems that shear rate less than 500 s-1 have shear thinning behavior, please verify!.
25- Line # 575-576, “This indicates that concentration of 1.0 wt.% PYNPs 575 at higher values of shear rate is considered as the best concentration that leads to a lower viscosity” the higher the polymer viscosity the better in controlling the mobility ration. Please revise the sentence.
26- Line # 588, Viscosity of HPAM at 5000 ppm should be included for comparison and to support you argument “The inclusion of NPs has improved the viscosity and viscoelastic properties of HPAM solution”
27- Please improve resolution of Figure 3a& 4a, scale not clear.
28- Line # 625. At CMC the particles a aggregates and therefore viscosity should increase not decrease. Which is not indicated by PYNPs nor by CSNPs. I suggest the author should remove this part from the paper and replace it with the paper "Thermal Stability", where author can add only thermal aging analysis figures.
29- To figure 15 and 16 please include the shear rate that used to measure viscosity and make sure the result is consistent with figure 11 and 12.
30- Figures 19-22 X-axis should be changed either to injected volume (cm3) or converted the axis’ values to pore volume by dividing the values by core’s pore volume. Also, I suggest removing the extrapolated lines from all figures and keep only the measured points.
31- Differential pressure and water cut curves should be added to figures 19-22.
32- HPAM adsorption is well documented, however, the static and dynamic adsorption of proposed nano polymer and the integration of HPAM and nano polymer should be addressed. That will determine potential propagation of polymer/nanoparticles in porous media as well as project economics.
33- Windsor type I, III irrelevant for polymer as the IFT is not low enough to increase oil recovery in this case, suggest remove this sentence from the conclusion and author should investigate other mechanisms that justify the incremental oil recovery (I.e improving volumetric sweep efficiency by log jamming effect, wettability alteration, mobility control….etc)
Author Response
Dear Milica Stanimirovic,
Greetings to you.
Thank you very much for your significant efforts and follow up. We have finished the required changes to our research work based on the reviewers' comments. In the list below our point to point response for the valuable notes suggested by the respected reviewers.
Thanks for your valuable efforts and please accept our best regards and appreciation.
Authors
Reviewer #2:
Dear Reviewer #2, Thank you very much for your wonderful comments which gave our work more strength. We did our best to answer all your questions and made the necessary changes based on your valuable comments. The changes are in tracked mode in the revised MS word manuscript.
Thank you in advance for your valuable efforts with us and please accept our best wishes.
Best regards,
Authors
Reviewer #2 comments:-
1- Line #33 please write was instead of is.
Response: Thank you for this valuable notice. In fact, we are not native English speakers, therefore, these errors happen. The necessary correction has been done.
2- Line # 59, “high number of works that are devoted to develop the bio-composites by blending such starch nanoparticles into the biopolymeric matrices”. Please cite those works and explain why you chose to integrate the starch nanoparticle with synthetic HPAM and not with biopolymer?
Response: Thank you for this valuable notice. Number of similar works have been cited (in color red) and added to the reference list as well. In fact, starch can be found in the stems, roots, fruits and seeds of many plants such as sweet potato, cassava, potato and many more. In addition to its original form, starch can be modified by reducing its size. Starch nanoparticles have small size and large active surface area, making them suitable for use as fillers or as a reinforcing material in biopolymers.
3- Line #87-88, 91and 100, please write “at the same time” instead of “in the same time” “improvement” instead of “improve” , ‘kinds of’ instead of ‘kinds on’ ‘as a function of’ instead of ‘as function of’
Response: Thank you for reminding about these spell errors. In fact, we are not native English speakers, therefore, these errors happen. The necessary corrections has been done.
4- Line # 104 “has resulted in obtaining the highest grafting percentage and water solubility, which were 1565.53 and 96.06%, respectively. Please revise the sentence and the numbers.
Response: Thank you for this valuable observation. The related sentence and numbers have been revised to the correct form.
5- Line # 113, I think polymer concentration should be chosen based on oil viscosity to control mobility and improve sweep efficiency, so what is the reason behind selecting 2000 ppm HPAM polymer?
Response: Thank you very much for this valuable observation. Yeah, it is true that chosen polymer concentration is related to the oil viscosity but in the current study the focus is not about HPAM polymer itself and besides to that we found many studies in literature have dealt before with HPAM at this concentration, therefore, we suggested to use this concentration and focus more on starch nano-polymer additives.
6- Line # 115, “optimum concentrations on the two types of nano polymer were selected based on contact angle” please explain? Refer to comment #5.
Response: Thank you for this important notice. In the main text (around line # 115) its mentioned that the optimum concentration ...............etc is estimated from interfacial tension measurements and rheological properties. Wettability presented by contact angle also has an important effect on selecting the best injected polymer. The less is the contact angle, the more is the injection is satisfactory but this is not clearly stated in the related paragraph.
7- Line # 145, please write Replace “composition of this oil after it was mixed with Fsol” instead of “composition for this oil after it mixed with Fsol”
Response: Thank you for this valuable suggested improvement. The related paragraph has been corrected according to the suggested one.
8- Line # I think oil viscosity at reservoir condition is important when designing a polymer flooding rather than using viscosity at 25 °C as reported in the methodology section.
Response: Thank you very much for this valuable note. All the necessary physical tests have been done at reservoir temperature 60 °C except for the paragraphs related to the synthesizing of nanoparticles which is done at 25 °C. If mentioned in the methodology that viscosity is estimated at 25 °C, this may be written by mistake.
9- Line # 136, since oil was solid at normal temperature, I believe it’s a waxy crude so wax content and pour point should be added to oil properties for better understanding to readers. Also, the impact of additive “Fsol” on oil viscosity should be reported.
Response: Thank you very much for this valuable note. Yeah, it is a waxy and heavy crude oil. The pour point and wax content for most Indonesian crude oils are laying between 35ºC - 40ºC and 20% - 25%, respectively. The Malaysian company that manufactured Fsol did not provide us with enough information about its content and viscosity, therefore, we were unable to reveal that.
10- It would be useful to include the composition of crude oil, core mineralology, and make-up brine for polymer solution since polymer behavior differs when it interacts with different types of rock and fluids.
Response: Thank you for this valuable notice. Concerning the mineralogy of the sandstone cores, some important information has been mentioned in table 1. The make-up brine has been made at salinity of 100 ppm to simulate for the original inspected oilfield which its formation water is very low in salinity.
11- Line # 148 and 152, “Partially hydrolyzed polyacrylamide (HPAM) 0.5% aqueous solution”, 0.5% is it w/w, w/v? Same for “Glacial acetic acid (CH3COOH) with purity of 99%”.
Response: Thank you for this valuable note. Regarding HPAM 0.5% it is based on w/v and regarding CH3COOH it is w/w basis. This info has been added into the related paragraphs.
12- Line # 153 “All chemicals” Which chemicals are referred to by "All chemicals" is not clear.
Response: Thank you very much for this valuable note. It is related to all chemicals manufactured by this company (QREC (Asia) Sdn. Bhd. Selangor, Malaysia). In order to not mislead with the chemicals used in the current study, the whole sentence that began with ''All chemicals......'' is deleted from this paragraph.
13- Line # 172, 174, Do you mean fiber or fibber ?
Response: Thank you very much for this valuable note. We ourselves confused with this word exactly! Thanks to your notice, we have checked some online dictionaries and the true word is fiber and not fibber. The right word has been inserted in the main text instead of the previous one.
14- Line # 199, 229 I think that Figure 1 and 2 are unclear, need to improve resolutions
Response: Thank you for valuable note. Figure 1 and 2 has been enlarged at a suitable margin to be more clarity for readers.
15- Line #203, The basis of selecting these values 2.5 mol/l acid concentration, 45°C and 5 days is unclear in the article, please elaborate.
Response: Thank you very much for this valuable note. An elaboration has been added in the main text about the reason behind selecting these parameters.
16- Line # 239 “geometrical shapes”! please explain
Response: Thank you for this valuable notice. An elaboration about the meaning of geometrical shapes has been added in the main text between brackets.
17- Line # 252 Please add citation to title of Table 3.
Response: Thank you very much for this valuable notice. A citation about table 3 is already inserted in the main text. On anyway, the same citation has been added at the end of title of table 3 according to request.
18- Line # 260 I think there is inconsistency of using 5000 ppm HPAM for viscosity measurement, 2000 ppm HPAM for coreflood. Please explain this inconsistency in the concentrations.
Response: Thank you very much for this valuable notice. The original concentration of HPAM that was supplied from the chemical company was 5000 ppm. In the rheological assessment of polymer formations, the actual used concentration was 2000 ppm as mentioned before in the response to point 5. In line # 260 it was written 5000 instead of 2000 ppm to refer to the original concentration that was received from the company. On anyway, this has been corrected in the related line by referring to the actual used concentration of 2000 and not 5000 ppm.
19- Line # 263: In situ shear rate for fluid flow inside reservoir far from wellbore around ~ 7-10 S-1, it is imperative to include low shear rates. As polymer solution behaviour, pertaining to the shear stress and shear rate, will help in determining the behaviour of the displacing fluids for EOR process, whether it is Newtonian or non-Newtonian.
Response: Thank you very much for this valuable note. The current study is not focusing about the dynamics of in situ shear rate inside the reservoir. In addition to that, the new studied region for the shear rate is now within 300 s-1 after taking in consideration all the notes about that.
20- Line # 246: CMC commonly used for surfactant not polymer that yields utra-low IFT. or polymers and nanoparticles, it is recommended to choose their concentrations based on their viscosity, injectivity, adsorption, and costs.
Response: Thank you very much for this valuable content. It is true that CMC aspect is used to calculate the ultra-low IFT for surfactants but it is used here for the hybrid polymers (HPAM/PYNPs and HPAM/CSNPs) as these polymer combinations are made from both industrial polymer (HPAM) and natural polymer (starches). Besides to that rheological assessment and adsorption test have performed for the two kinds of nano-polymers and both of them were satisfactory. We did not include these details in the current work. The adsorption test for the hybrids was performed based on calculating a parameter called Resistant Factor (RF) which is the relation between pressure difference for the hybrids divided by the pressure difference for brine and this parameter should be less than 1.2. It was calculated that RF was 0.403 for the PYNP hybrid and 0.35 for CSNP solution at tested concentrations. The injectivity test has been already done in conjunction with the adsorption test.
21- Line # 303, 315, 320, I think there is also inconsistency in flow rates used for oil flood (8 cm3 per minute), brine flood (5 cm3/minute) and polymer (3 to 3.5 cm3/minute). Please explain
Response: Thank you very much for this valuable question. Different liquids with different viscosities lead to different load to be applied from the Teledyne pump. Therefore, it is hard to consolidate the flow rate for a single value as that may make injection time for some liquids fast and for others slow.
22- Line #333, 90.53 and 85%, w/w.% or v/v%?
Response: Thank you for this valuable note. They are both weight based (i.e w/w%) and for organization matter we did not show that clearly next to these numerical values.
23- Line # 340 “The average particle size for PYNPs and CSNPs based on intensity percent (which is useful in detecting small amounts of aggregation) was 363.12 and 52.92 nm, respectively” this sentence contradicts with zeta potential results -36.3 and -10.7 for PYNPs and CSNPs, please explain.
Response: Thank you for this very important note. Particle size distribution (that is performed in industry research laboratory of UTM) is based on three major major variables and methods which they are intensity, volume and number percent. The one that is sensitive in detecting any agglomeration of nanoparticles (which is our focus) is based on intensity percent measurement (the first one). Therefore, we only considered the results obtained by this analysis and calculated the average particle size and did not take into account the results obtained by the other two methods. Concerning zeta potential results, the numbers displaced are in millivolts (mv) and not in w/w% like particle size and as explained in table 3, the big are those numbers (in positive or negative) the more stable are the particles especially if the numbers exceed +30 or -30 mv.
24- Line #571, “From these figures, the viscosity increased with increasing shear rate for both PYNPs and CSNPs” it seems that shear rate less than 500 s-1 have shear thinning behavior, please verify!.
Response: Thank you very much for this valuable question. Shear thinning behavior is the non-Newtonian behavior of fluids whose viscosity decreases with increasing shear rate. In fact, this has not achieved for all tested concentrations in our work. At some concentrations, the viscosity increased with increase shear rate, therefore, this is considered as shear thickening behavior while for other concentrations the viscosity decreased with increasing shear rate (shear thinning behavior). Since shear thickening behavior happens as noticed, therefore, for shear rate less than 500 s-1 the hybrids laid between shear thinning to shear thickening behavior.
25- Line # 575-576, “This indicates that concentration of 1.0 wt.% PYNPs 575 at higher values of shear rate is considered as the best concentration that leads to a lower viscosity” the higher the polymer viscosity the better in controlling the mobility ration. Please revise the sentence.
Response: Thank you for this valuable information. The said sentence has been revised to the following:
"An intermediate concentration of 1.0 wt.% PYNPs is considered as the best concentration that led to a lower viscosity from the rheological measurements"
26- Line # 588, Viscosity of HPAM at 5000 ppm should be included for comparison and to support you argument “The inclusion of NPs has improved the viscosity and viscoelastic properties of HPAM solution”
Response: Thank you very much for this precious notice. We really appreciate it. Our argument has been supported by comparison our work with another study performed by Rellegadla et al. [10] which they depended only on HPAM solution to increase oil recovery. On anyway, it was managed by our team to not perform injectivity test with HAPM as this has been done by many researchers.
27- Please improve resolution of Figure 3a& 4a, scale not clear.
Response: Thank you for this valuable observation. The resolution of figures 3a & 4a has been improved and besides to that the size of figures has been increased little bit to show more details.
28- Line # 625. At CMC the particles a aggregates and therefore viscosity should increase not decrease. Which is not indicated by PYNPs nor by CSNPs. I suggest the author should remove this part from the paper and replace it with the paper "Thermal Stability", where author can add only thermal aging analysis figures.
Response: Thank you very much for this valuable notice. It is not evidence for us that the nanoparticles did not aggregate at certain concentration. It was shown from zeta potential analysis that the stability of PYNPs (-36.3 mv) is much better than CSNPs (-10.7 mv) and as a result the performance of PYNPs solution was better in recovery oil than the other nano-solution. On anyway, much improvement has been done on the paragraph related to the rheological properties of PYNPs & CSNPs and the focus was increased on low shear rates only. In this regard, figures 11 and 12 have been deleted and more elaboration has been highlighted on thermal aging as shown in paragraph 3.6. The authors hope that the modifications made in this regard give some answers for the reviewers.
29- To figure 15 and 16 please include the shear rate that used to measure viscosity and make sure the result is consistent with figure 11 and 12.
Response: Thank you very much for this valuable notice. For figures 15 & 16 we did not record the value of shear rate during measuring the viscosity. This is because we focused only on the aging time and wanted to confirm the stability of the polymer solution at the inspected concentration. Concerning figures 11 & 12 they are deleted as explained before because they did not give sufficient information according to the point of view of all reviewers.
30- Figures 19-22 X-axis should be changed either to injected volume (cm3) or converted the axis’ values to pore volume by dividing the values by core’s pore volume. Also, I suggest removing the extrapolated lines from all figures and keep only the measured points.
Response: Thank you very much for this valuable note. We have changed the x-axis for all figures to the injected volume (cm3) and we preferred to maintain the extrapolated lines in the same figures in order to distinguish and separate between different types of injection.
31- Differential pressure and water cut curves should be added to figures 19-22.
Response: Thank you for the important notice. Water flooding regions has been located for all figures 19-22. We focused in our study in obtaining the recovery factor (RF) against the injected volume, therefore, we did not draw curves that relate differential pressure with water cut. The pressure difference has been calculated in the adsorption test to evaluate the hybrid solutions.
32- HPAM adsorption is well documented, however, the static and dynamic adsorption of proposed nano polymer and the integration of HPAM and nano polymer should be addressed. That will determine potential propagation of polymer/nanoparticles in porous media as well as project economics.
Response: Thank you very much for this valuable notice. This is exactly in our intension and we have agreed to address this in our next paper. In fact, static and dynamic adsorption of proposed nano-polymers is something new and important.
33- Windsor type I, III irrelevant for polymer as the IFT is not low enough to increase oil recovery in this case, suggest remove this sentence from the conclusion and author should investigate other mechanisms that justify the incremental oil recovery (I.e improving volumetric sweep efficiency by log jamming effect, wettability alteration, mobility control….etc)
Response: Thank you for this valuable note. We did not elaborate much about the phase behavior and Windsor types. As it is known Windsor type changes from type I to type III and to type II when the salinity increases accordingly. Since we have worked with a formation water that has salinity of only 100 ppm, therefore, Winsor type I is likely to happen for the most injections done in this work. Even though, it was possible to capture Windsor type III for the solution that contains PYNPs which gave some confirmation about the suitability of this starch nanoparticle to be involved in future EOR operations since it is cheap and abundant in nature. The ultra-low IFT (magnitude of 10-3 mN/m) cannot reached depending only on the kind of surfactant and special preparations are needed to reach to this level. In the same time, the ultra-low IFT cannot be always obtained by relying only on the surfactant concentration. Hope this clarifies the situation.
Round 2
Reviewer 1 Report
I believe that the authors simply do not understand the meaning of such terms as confidence limit, experimental error and so on.
Then they did not change presentation of Figs 15 and 16/
This is a bad manner to change experimental oints by sectiond of straight lines instead of drawing averaging smooth cirves.
Please, make necessary corrections herer and be accurate in your further presentations.
Author Response
Your notes are highly appreciated and valued. Yeah, it is true that we did not understand what you have mentioned in the first round. On anyway, figs 15 and 16 have been redrawn using averaging smooth curves as requested. In doing so, two approaches have been adopted in drawing the averaging smooth: Savitzky-Golay and Adjacent-Averaging.
Warm wishes and best regards,
Hasanain
Reviewer 2 Report
Dear respected Editor.
The author has made a litle improve to the paper. Some critical issues have not yet been resolved. The author claims nano particles improve polymer performance, but polymer performance has not been tested. Its imperative to include polymer performance with and without nanoparticles Also, a pressure drop curve should be added to the oil recovery chart to confirm that nanoparticles do not damage the core. In addition, viscosity measurement should include shear rates that are within the real and practical values inside reservoirs.
Best Regards,
Abdelazim,
Author Response
Your excellent notes are deeply appreciated. We made many improvements to our manuscript. Polymer performance using HPAM alone has been evaluated for injection as shown in fig.24 and the results have been compared with adding the two nanoparticles as requested. Pressure drop curve versus injected volume has been established for the two nanoparticles besides for water flooding as shown in fig. 23 and the results are discussed. Also, obtained shear rate values have been examined for practical and real values inside the reservoirs.
Warm regards and best wishes,
Hasanain